# Identification of Accurate Reference Genes for qRT-PCR Analysis of Gene Expression in *Eremochloa ophiuroides* under Multiple Stresses of Phosphorus Deficiency and/or Aluminum Toxicity

**DOI:** 10.3390/plants12213751

**Published:** 2023-11-02

**Authors:** Ying Chen, Qingqing He, Xiaohui Li, Yuan Zhang, Jianjian Li, Ling Zhang, Xiang Yao, Xueli Zhang, Chuanqiang Liu, Haoran Wang

**Affiliations:** 1State Key Laboratory of Tree Genetics and Breeding, Ministry of Education of China, Co-Innovation Center for the Sustainable Forestry in Southern China, Nanjing Forestry University, Nanjing 210037, China; ychen@njfu.edu.cn (Y.C.);; 2The National Forestry and Grassland Administration Engineering Research Center for Germplasm Innovation and Utilization of Warm-Season Turfgrasses, Institute of Botany, Jiangsu Province and Chinese Academy of Sciences, Nanjing Botanical Garden, Mem. Sun Yat-Sen, Nanjing 210014, China; 3Jiangsu Key Laboratory for the Research and Utilization of Plant Resources, Institute of Botany, Jiangsu Province and Chinese Academy of Sciences, Nanjing Botanical Garden, Mem. Sun Yat-Sen, Nanjing 210014, China

**Keywords:** *Eremochloa ophiuroides*, qRT-PCR, reference gene, phosphorus deficiency, aluminum toxicity, acid soil adversity

## Abstract

Centipedegrass (*Eremochloa ophiuroides* (Munro.) Hack.) is a species originating in China and is an excellent warm-season turfgrass. As a native species in southern China, it is naturally distributed in the phosphorus-deficient and aluminum-toxic acid soil areas. It is important to research the molecular mechanism of centipedegrass responses to phosphorus-deficiency and/or aluminum-toxicity stress. Quantitative Real-Time PCR (qRT-PCR) is a common method for gene expression analysis, and the accuracy of qRT-PCR results depends heavily on the stability of internal reference genes. However, there are still no reported stable and effective reference genes for qRT-PCR analysis of target genes under the acid-soil-related stresses in different organs of centipedegrass. For scientific rigor, the gene used as a reference for any plant species and/or any stress conditions should be first systematically screened and evaluated. This study is the first to provide a group of reliable reference genes to quantify the expression levels of functional genes of *Eremochloa ophiuroides* under multiple stresses of P deficiency and/or aluminum toxicity. In this study, centipedegrass seedlings of the acid-soil-resistant strain ‘E041’ and acid-soil-sensitive strain ‘E089’ were used for qRT-PCR analysis. A total of 11 candidate reference genes (*ACT*, *TUB*, *GAPDH*, *TIP41*, *CACS*, *HNR*, *EP*, *EF1α*, *EIF4α*, *PP2A* and *actin*) were detected by qRT-PCR technology, and the stability of candidate genes was evaluated with the combination of four internal stability analysis software programs. The candidate reference genes exhibited differential stability of expression in roots, stems and leaves under phosphorus-deficiency and/or aluminum-toxicity stress. On the whole, the results showed that *GAPDH*, *TIP41* and *HNR* were the most stable in the total of samples. In addition, for different tissues under various stresses, the selected reference genes were also different. *CACS* and *PP2A* were identified as two stable reference genes in roots through all three stress treatments (phosphate deficiency, aluminum toxicity, and the multiple stress treatment of aluminum toxicity and phosphate deficiency). Moreover, *CACS* was also stable as a reference gene in roots under each treatment (phosphate deficiency, aluminum toxicity, or multiple stresses of aluminum toxicity and phosphate deficiency). In stems under all three stress treatments, *GAPDH* and *EIF4α* were the most stable reference genes; for leaves, *PP2A* and *TIP41* showed the two highest rankings in all three stress treatments. Finally, qRT-PCR analysis of the expression patterns of the target gene *ALMT1* was performed to verify the selected reference genes. The application of the reference genes identified as internal controls for qRT-PCR analysis will enable accurate analysis of the target gene expression levels and expression patterns in centipedegrass under acid-soil-related stresses.

## 1. Introduction

Quantitative Real-Time PCR (qRT-PCR) is a widely used technology for nucleic acid quantitative analysis of target gene expression, which has the characteristics of high sensitivity, large dynamic range, high detection efficiency and accurate quantification [1,2,3,4]. The result of qRT-PCR is commonly influenced by RNA quality, the efficiency of reverse transcription and PCR amplification. To obtain the real differences in specific expression of target genes, therefore, it is necessary to correct and standardize the data using stably expressed internal reference genes [5,6]. The accuracy of qRT-PCR results depends largely on the stability of internal reference genes [7,8].

Theoretically, the internal reference gene should have a stable expression pattern in various experimental samples [8,9,10]. Several studies reported organ-specific or stress-related stable reference genes; however, the reported reference genes are not consistently expressed in different tissues or environmental conditions. For example, in *Cunninghamia lanceolata*, the most stable internal reference gene was *actin* in the root, but *GAPDH* in the cotyledon [4]; in tomato, the expression of *EF1α* was stable under nitrogen stress and low-temperature stress, but unstable under light stress [9]. Obviously, there is no one gene which can be used as a universal reference for all plant species. Considering the complex genetic networks regulating plant responses across diverse environmental conditions and growth stages, selecting suitable reference genes for specific plant species, organs and environments is essential for precise quantification of gene expression levels.

Centipedegrass (*Eremochloa ophiuroides* (Munro.) Hack.) is a species originating in China and usually used as an excellent warm-season turfgrass [11]. As a native species in southern China, centipedegrass is naturally distributed in the acid soil areas, with phosphorus-deficiency and aluminum-toxicity stress. It is essential to study the molecular mechanisms of adaptation to acid soil conditions in centipedegrass. This is to guide the breeding of P-efficient and/or aluminum-tolerant varieties for improving the ecology and utilization rate of acid soil barren land areas. To detect stress-responsive genes and determine the molecular mechanisms that regulate centipedegrass tolerance to these acid soil-related stresses, qRT-PCR can be effective for the examination and quantification of target gene expression levels and patterns. However, in previous studies, the reference genes under P-deficiency and/or aluminum-toxicity stress have not been selected and validated in centipedegrass.

For scientific rigor, there is no one gene that can be used as a reliable reference for any plant species and/or any stress conditions without systematic screening and evaluation. For the purpose of studying target gene expression patterns with qRT-PCR in the leaves, stems and roots of centipedegrass of acid soil-resistant and acid soil-sensitive genotypes in response to phosphorus-deficiency and/or aluminum-toxicity stress, the systematic selection of relevant reference genes would be scientifically valuable. A total of 11 potential reference genes were selected, including *ACT (Actin)*, *TUB (Tubulin)*, *GAPDH (Glyceroldehyde-3-phosphate dehydrogenase)*, *TIP41 (Tonoplast intrinsic protein 41)*, *CACS (Clathrin adaptor complexes)*, *HNR (Heterogeneous nuclear ribonucleoprotein)*, *EP (Expressed protein)*, *EF1α (Elongation factors 1 alpha)*, *EIF4α (Eukaryotic translation initiation factor 4 alpha)*, *PP2A (Serine/threonine protein phosphatase)* from centipedegrass transcriptome data (RNA-seq), and another “*actin*” reported in previous studies [12]. A total of four algorithms, including geNorm, NormFinder, BestKeeper and RefFinder, were used to comprehensively evaluate the expression stability of these candidate reference genes. Finally, the selected reference genes were verified by examining the expression pattern of a stress-responsive *ALMT1* gene. This study is the first to provide a group of reliable reference genes for quantifying the expression levels of functional genes of *Eremochloa ophiuroides* under multiple P-deficiency and/or aluminum-toxicity stresses.

## 2. Results

### 2.1. Gene-Specific Amplification Efficiency of Candidate Reference Genes

The specificities of primer pairs of 10 candidate reference genes selected from centipedegrass transcriptome data were confirmed by agarose gel electrophoresis and melting curves from qRT-PCR, which revealed a single DNA band in each gel lane or single peak from each amplicon (Appendix A). In addition, the primer pair of *actin* has been reported in previous studies [12]. Using the LinRegPCR version_2014.x software [13], we calculated the mean amplification efficiency of the 11 pairs of primers through the data acquired from qRT-PCR experiments, and the mean amplification efficiencies of all primers ranged from 1.843 (*GAPDH*) to 2.000 (*EP*) (Table 1), with an ideal value in the range 1.8 ≤ E ≤ 2 [14].

### 2.2. Expression Levels and Variations of Candidate Reference Genes

The Ct values of the candidate reference genes were assayed through qRT-PCR experiments. The Ct values were found in a wide range (from 14.80 to 27.18) across all samples, but the values for most of the genes were between 20 and 24. As the Ct values are inversely proportional to the mRNA transcript levels, *EF1α* showed the highest expression level (mean Ct of 16.70), while *actin* exhibited the lowest expression level (mean Ct of 23.43). Moreover, a lower variation in Ct values indicates less variability of the gene in the expression levels during qRT-PCR analysis. *PP2A* showed the least variation with 3.56 Ct (19.91–23.47), and *TUB* exhibited the highest variation with 7.44 Ct (16.17–23.61) (Figure 1).

### 2.3. geNorm Analysis

Based on the geNorm analysis (geNorm_v3.5), the ranking of candidate reference genes for samples of root, stem and leaf under three diverse stress treatments are shown by M values (Figure 2A and Appendix A). A lower M value represents a higher stability of the reference gene (the default threshold of M was less than 1.5). Over all samples, *GAPDH* and *TIP41* were the most stable genes, followed by *PP2A*, *CACS*, *HNR*, *EIF4α*, *EP*, *EF1α*, *ACT*, and *TUB*, while *actin* was the least stable gene. Meanwhile, we realized that the ranking of candidate reference genes would vary in different tissues under multiple stress conditions.

In P-deficiency-treated roots (PR), *CACS* and *EF1α* were the most stable genes. In Al toxicity-treated roots (AR), *EF1α* and *EIF4α* were the most stable genes. In Al-toxicity and P-deficiency multi-stress-treated roots (MR), *EP* and *PP2A* were the most stable genes. For the root samples pooled from all three stress treatments (WR), *EP* and *PP2A* were the most stable genes.

In P-deficiency-treated stems (PS), *EP* and *EIF4α* were the most stable genes. In Al-toxicity-treated stems (AS), *CACS*, *PP2A*, *GAPDH* and *ACT* were the most stable genes. In Al-toxicity and P-deficiency multi-stress-treated stems (MS), *GAPDH*, *HNR* and *EF1α* were the most stable genes. For the stem samples pooled from all three stress treatments (WS), *EP* and *EIF4α* were the most stable genes.

In P-deficiency-treated leaves (PL) and Al-toxicity-treated leaves (AL), *GAPDH* and *TIP41* were the most stable genes. In Al-toxicity-treated leaves (AL), *ACT* and *TUB* were the most stable genes. In Al-toxicity and P-deficiency multi-stress-treated leaves (ML), *TIP41* and *CACS* were the most stable genes. *GAPDH* and *TIP41* were the most stable genes for the leaf samples pooled from all three stress treatments (WL).

For the normalization of target gene expression levels with qRT-PCR, the optimal number of reference genes required was determined by pair variation with a threshold of 0.15. If the Vn/Vn + 1 value is less than 0.15, the optimal number of reference genes is N [15]. In this study, the V2/3 values in the WR, PR, AR, MR, PS, PL, AL and ML samples were below 0.15 (Figure 2B), indicating two reference genes are required for normalization. The V3/4 value of 0.138 shows that three genes are sufficient for normalization of the MS sample. The V4/5 (0.138) of the AS sample indicates that four reference genes are required. However, the threshold value of 0.15 should not be considered as inflexible as several reports have reported higher cutoff values of Vn/n + 1 [16,17,18,19]. Here, V2/3 (0.160) and V3/4 (0.165) of the WL sample exhibited small variations, indicating two reference genes should be sufficient for normalization. Similarly, a small variation between V2/3 (0.228), V3/4 (0.247) and V4/5 (0.223) of the WS sample would suggest that the addition of a third gene has no significant effects on the normalization factor.

### 2.4. NormFinder Analysis

The expression stability of the candidate reference genes was also calculated by NormFinder_v20 (Appendix A). In all the samples, *GAPDH* was the most stable gene, followed by *HNR*, *EIF4α*, *CACS*, *TIP41*, *PP2A*, *ACT*, *EF1α*, *EP*, and *TUB*, while *actin* was the least stable gene. Similar to the results of geNorm, the ranking of candidate reference genes varied in different tissues under multiple-stress conditions. *CACS* was the most stable gene in the three samples of root tissue (WR, AR and MR), while *GAPDH* was the most stable gene in three samples (WS, AS and MS) of stem tissue and the AL sample of leaf tissue. In addition, *PP2A* was revealed as the most stable reference gene in the WL samples, while *HNR* in the PL sample, *actin* in the ML sample, *EF1α* in the PR sample and *TIP41* in the PS sample.

### 2.5. BestKeeper Analysis

The stability of reference genes was also calculated by BestKeeper_v1 based on the standard deviation (SD), correlation coefficient (R) and coefficient of variation (CV) of Ct values. In general, the SD value is inversely related to the stability of the internal reference gene, and a gene with an SD greater than 1 would be unstable. The ranking of reference genes by BestKeeper analysis is shown in Appendix A. For the root tissue, *CACS* was identified as the most stable gene in the WR and AR samples, consistent with the results from the NormFinder analysis, while the results showed *TIP41* in the PR sample and *ACT* in the MR sample as the most stable genes. In the WS and MS samples, *actin* was identified as the most stable gene by the BestKeeper analysis. But, it ranked ninth with geNorm for WS and MS. With NormFinder, it ranked eighth and ninth in WS and MS, respectively. Also, the most stable gene identified in the PS and AS samples by the BestKeeper analysis showed differences from the results of the geNorm and NormFinder analyses, which chose *ACT* for the PS sample and *PP2A* for the AS sample. For the leaf tissue, *TIP41* was identified as the most stable gene in the WL and PL samples, which ranked second. This gene also ranked second with geNorm for WL and PL, respectively, and fourth and seventh, respectively, with NormFinder. In the AL and ML samples, *EP* was revealed as the most stable reference gene, which with geNorm, ranked ninth and sixth, respectively, and tenth and second, respectively, with NormFinder.

### 2.6. Comprehensive Ranking by RefFinder Analysis

Finally, combining the geNorm, NormFinder and BestKeeper outputs, RefFinder (http://blooge.cn/RefFinder/; accessed on 1 October 2023) was used to generate a comprehensive ranking of the candidate reference genes (Table 2). Based on the RefFinder analysis (Table 3), *CACS* and *PP2A* were identified as two stable reference genes in roots through all three stress treatments (WR sample), while *actin* was the least stable gene, which was consistent with the results of the geNorm and BestKeeper analyses. Moreover, *CACS* was also stable as a reference gene in roots under phosphate deficiency (PR sample), aluminum toxicity (AR sample), or combined aluminum toxicity and phosphate deficiency (MR sample). In stems under all three stress treatments (WS sample), *GAPDH* and *EIF4α* were found to be the most stable reference genes. *TIP41* and *ACT* were relatively stable for the PS sample, and *GAPDH* was the most stable reference gene in the MS sample. *GAPDH*, *CACS*, *PP2A* and *EIF4α* were four stable reference genes in the AS sample. In addition, *TUB* was the most unstable reference gene in the stems of all samples. For leaves in all three stress treatments (WL sample), *PP2A* and *TIP41* showed the two highest rankings, while *EF1α* was shown to be the least stable. *GAPDH* was relatively stable for the PL and AL samples, *TIP41* was relatively stable for the WL, PL and ML samples, and *HNR* was relatively stable for the AL sample. In addition, *CACS* was the most stable reference gene in the ML sample.

### 2.7. Validation of Candidate Reference Genes

Through RefFinder, we could select the most stable and unstable candidate internal reference genes in different tissues under multiple-stress conditions. In order to verify the practicality of these internal reference genes, we used *ALMT1* as the target gene and selected genes as the internal reference genes for verification. *ALMT1* genes have been implicated in malate transport in response to aluminum in *Triticum aestivum* (*TaALMT1*) [20], *Brassica napus* (*BnALMT1*) [21] and *Arabidopsis thaliana* (*AtALMT1*) [22]. Moreover, *ALMT1* is also important in the root developmental response to phosphate availability [23]. To confirm the utility of the selected reference genes, the expression patterns of *ALMT1* in response to phosphate deficiency (PR sample), aluminum toxicity (AR sample), and combined aluminum toxicity and phosphate deficiency (MR sample) were examined, in the acid-soil-resistant strain ‘E041’ and the acid-soil-sensitive strain ‘E089’, respectively. The two most stable reference genes (*CACS* and *PP2A*) and the least stable reference gene (*actin*) were selected for the validation test. With the most stable reference genes (*CACS* and *PP2A*) analyzed either alone or in combination, similar expression patterns could be observed. However, the expression of *ALMT1* failed to exhibit a consistent pattern when using the least stable “*actin*” as the reference gene (Figure 3).

## 3. Discussion

With the characteristics of rapidness, high sensitivity, accuracy, and reproducibility, qRT-PCR has been widely used to research gene expression [1,2,3,4]. However, the application of improper reference genes can lead to incorrect experimental conclusions. To obtain accurate qRT-PCR analysis results of gene expression, it is important to select appropriate reference genes to standardize the data. However, there are no previous reports providing a systematic evaluation of reference genes of *Eremochloa ophiuroides* for qRT-PCR analysis.

In this work, 11 candidate reference genes were selected: *ACT*, *TUB*, *GAPDH*, *TIP41*, *CACS*, *CACS*, *HNR*, *EP*, *EIF4α*, *PP2A*, and *actin*. Through analysis based on Ct values, the average Ct values of the candidate reference genes were found to vary from 16.70 (*EF1α*) to 23.43 (*actin*). *PP2A* showed the least variation with 3.56 Ct (19.91–23.47), *TUB* exhibited the highest variation with 7.44 Ct (16.17–23.61) (Figure 1). According to previous studies, genes with low expression levels or high expression variation may not be appropriate for data normalization, and multiple tools must be used to evaluate the reference genes comprehensively [24,25].

To date, several algorithms have been used for selecting and calculating reference gene stability. In this study, multiple algorithms (NormFinder, geNorm, BestKeeper and RefFinder) were applied to analyze the stability of candidate reference genes in samples of different tissues and/or under diverse abiotic stress treatments in *Eremochloa ophiuroides*. We found that the most stable five genes calculated by the four algorithms were partially similar. For instance, in all the samples, the most stable five genes of the multiple algorithms always included *GAPDH*, *TIP41*, and *HNR*. In addition, *CACS* was selected by GeNorm, NormFinder and RefFinder; *PP2A* was selected by GeNorm, BestKeeper, and RefFinder; *EIF4α* was selected by NormFinder; and *ACT* was selected by BestKeeper.

However, some different stability analysis results were found using the diverse algorithms. For example, for all the samples of this study, GeNorm analysis indicated that *GAPDH*, *TIP41* and *PP2A* were the genes with the highest stability; NormFinder showed that the *GAPDH*, *HNR* and *EIF4α* genes had the highest stability; and *PP2A*, *TIP41*, *ACT* were thought to be the most stable genes by BestKeeper. This phenomenon was also observed in the study of reference genes in other plant species, such as *Cryptomeria fortunei* [26,27], *Passiflora edulis* [28], and *Liriodendron chinense* [29]. This is probably because each algorithm has different methods and principles for evaluating stability, and the conclusions are, therefore, diverse. Finally, the synthetic analysis by RefFinder, which combines the geNorm, NormFinder and BestKeeper outputs, showed that *GAPDH* was the most stable gene for all the samples of this study, while *actin* was the least stable.

As the most commonly used housekeeper genes, *GAPDH* and *actin* have all been reported as endogenous controls for quantitative RT-PCR analysis in previous studies [30,31]. In this study, for all the samples, *GAPDH* ranked first among the three algorithms (NormFinder, geNorm, RefFinder) and fourth in BestKeeper. This result is similar to findings of previous studies in plants such as creeping bentgrass [32], and even in animals, including humans [33,34]. However, in this study, the frequently used reference gene *actin* was the most unstable gene for all samples. This seems inconsistent with some previous studies, but there are still other studies with findings similar to ours; for instance, *actin* was shown to not be a reliable reference gene for studies assessing the effect of ocean acidification on *Emiliania huxleyi* [35]. Thus, there is no gene which could be used as a universal reference for all plant species. The selected reference genes of this study should be valuable, for the purpose of quantifying target gene expression with qRT-PCR in the leaves, stems and roots of centipedegrass with acid-soil-resistant and acid-soil-sensitive genotypes in response to phosphorus-deficiency and/or aluminum-toxicity stress.

## 4. Materials and Methods

### 4.1. Plant Materials and Stress Treatments

Centipedegrass seedlings of the acid-soil-resistant strain ‘E041’ and acid-soil-sensitive strain ‘E089’ were used for qRT-PCR analysis. The plant materials were gathered from the Turfgrass Germplasm Resource nursery at the Institute of Botany, Chinese Academy of Sciences, Jiangsu Province. The stolons were taken from the nursery, cut into 4~5 cm pieces that included two nodes and cultivated hydroponically. Based on the centipedegrass hydroponic culture system [36], the hydroponic seedlings were pre-cultured to the best growth state (for about 2 weeks), and treated subsequently using the “long-term Al-P alternate treatment” method [37]. Four cultivation and treatment conditions, denoted as ‘Control’ (−Al/+Pi), ‘P-deficiency’ (−Al/−Pi), ‘Al-toxicity’ (+Al/+Pi), and ‘Al-toxicity and P-deficiency’ (+Al/−Pi), were set up as follows:

Control (−Al/+Pi): The set up consisted of alternating treatments of a normal nutrient solution (1/2 Hoagland culture solution, in which the concentration of Pi was 500 μM, pH 4.0) and a Mock solution (aqueous solution of 0.5 mM CaCl_2_, pH 4.0).

P-deficiency (−Al/−Pi): The set up consisted of alternating treatments of a P-deficient solution (1/2 Hoagland culture solution, with the concentration of Pi adjusted to 10 μM, pH 4.0) and a Mock solution (aqueous solution of 0.5 mM CaCl_2_, pH 4.0).

Al-toxicity (+Al/+Pi): The set up included alternating treatments of a normal nutrient solution (1/2 Hoagland culture solution, in which the concentration of Pi was 500 μM, pH 4.0) and an Al-toxicity solution (aqueous solution of 0.5 mM CaCl_2_ with 1.5 mM AlCl_3_, pH 4.0).

Al-toxicity and P-deficiency (+Al/−Pi): The set up comprised alternating treatments of a P-deficient solution (1/2 Hoagland culture solution, with the concentration of Pi adjusted to 10 μM, pH 4.0) and an Al-toxicity solution (aqueous solution of 0.5 mM CaCl_2_ with 1.5 mM AlCl_3_, pH 4.0).

After 2 weeks of growing in the 1/2 Hoagland culture solution with normal nutrient, the seedlings were exposed to an aqueous solution of 0.5 mM CaCl_2_, either with (+Al) or without (−Al) AlCl_3_, for 1 day. Subsequently, they were grown in the normal nutrient (+Pi) or P-deficiency (−Pi) culture solutions on alternate days. After 3 weeks of alternating treatments, the leaf, stem and root tissues were harvested (3 biological replicates). The fresh samples were cleaned with ddH_2_O and frozen in liquid nitrogen immediately.

### 4.2. RNA Isolation and cDNA Synthesis

Total RNA was extracted from each sample using the RNAprep Pure Plant Kit (Tiangen) with on-column DNaseI digestion. The nucleic acid concentration was quantified by an a NanoDrop 2000 spectrophotometer (Thermo Scientific, Wilmington, NC, USA) at 260 nm. The 260/280 nm ratio was maintained within the range of 1.80~2.20, and the 260/230 nm ratio was maintained at approximately 2.00. The cDNA was synthesized from 1 µg of total RNA using the Promega ImPromII Reverse Transcription System (Promega, Madison, WI, USA) with Oligo(dT)_18_ primer.

### 4.3. Selection of Candidate Reference Genes and Primer Design

According to the reference genes reported in other species (*Cynodon dactylon*, *Agrostis stolonifera*, *Oryza sativa*, *Sorghum bicolor*) [19,32,38,39], ten homologues genes were identified from the transcriptome data of centipedegrass (RNA-seq) and selected as candidate reference genes, including *ACT*, *TUB*, *GAPDH*, *TIP41*, *CACS*, *HNR*, *EP*, *EF1α*, *EIF4α* and *PP2A*. In addition, we selected another ‘*actin*’ gene as a candidate gene which was used in previous studies on centipedegrass root development [12].

Primers for the candidate reference genes (*ACT*, *TUB*, *GAPDH*, *TIP41*, *CACS*, *HNR*, *EP*, *EF1α*, *EIF4α*, and *PP2A*) were designed using the software PRIMER PREMIER version_5.0 with the following parameters: primer length of 21~25 bp, GC content of 40~60%, a melting temperature of 55~65 °C, and an amplicon length of 120~288 bp (Table 1). The mean amplification efficiencies of the primers were calculated with LinRegPCR version_2014.x software [13].

### 4.4. qRT-PCR Analysis

qRT-PCR was performed on an ABI ViiA 7 platform (ABI, Foster City, CA, USA) and in a 20 μL reaction system of the FastStart Universal SYBR Green Master Mix (Roche Applied Science, Indianapolis, IN, USA). The thermal cycling program was as follows: 50 °C for 2 min, 95 °C for 10 min, and 40 cycles of 95 °C for 15 s, and 60 °C for 1 min. This was followed by a melt curve step at 95 °C for 15 s, and 60 °C for 1 min. All reactions were performed in three replicates.

### 4.5. Stability of Internal Reference Genes

The expression stabilities of the 11 candidate reference genes were calculated using the four analysis tools: geNorm [15], NormFinder [40], BestKeeper [41], and RefFinder [42].

GeNorm identifies the stability of a set of tested reference genes in a given sample set under certain experimental conditions and calculates the normalization factor of gene expression by geometric averaging of all the reference genes [15]. GeNorm evaluates the stability of gene expression by the M value. A lower M value represents a higher stability of gene expression. According to the pair variation with a threshold of 0.15, the optimal number of reference genes is determined. If the Vn/Vn + 1 value is less than 0.15, the optimal number of reference genes is N, that is, no additional genes are required for standardization. However, if the Vn/Vn + 1 value is greater than 0.15, the optimal number of reference genes is N + 1 [15].

NormFinder is a mathematical model-based gene expression variation estimation method that identifies the best standardized gene in a set of candidate genes based on the size of the stable value [40].

BestKeeper used the standard deviation (SD), coefficient of variation (CV) and correlation coefficient (R) of Ct values to calculate the stability of candidate reference genes. The greater the correlation coefficient, the lower the standard deviation (SD) and the lower the coefficient of variation (CV), the higher the stability of the internal reference gene. When SD > 1, the expression of the internal reference gene is unstable [41].

RefFinder was developed for comprehensive evaluation and screening of reference genes from extensive experimental datasets. It synthesizes the results of the currently available major analysis tools (geNorm, Normfinder, BestKeeper, and deltaCt) to compare and rank the tested candidate reference genes [42].

### 4.6. Validation of Reference Gene Expression

To test the selected reference genes, we used *ALMT1* as a target gene and investigated the expression pattern of the two most stable and the one most unstable reference genes in various conditions. The primers of *ALMT1* for qRT-PCR were 5′-ACTACCCAGTCTACCCAATCTAAAC-3′ (forward) and 5′-GCTGCCTGAGGTTGATGAAT-3′ (reverse). All further steps are described above.

## 5. Conclusions

In this study, we analyzed the expression levels of eleven potential reference genes (*ACT*, *TUB*, *GAPDH*, *TIP41*, *CACS*, *HNR*, *EP*, *EF1α*, *EIF4α*, *PP2A* and *actin*) in the leaves, stems and roots of centipedegrass with acid-soil-resistant and acid-soil-sensitive genotypes in response to phosphorus-deficiency and/or aluminum-toxicity stress. Then, a total of four algorithms, including geNorm, NormFinder, BestKeeper and RefFinder, were used to comprehensively evaluate the expression stability of these candidate reference genes. These analyses indicate that diverse genes should be used as the reference under all the experimental conditions.

Based on the comprehensive final ranking of the candidate reference genes (Table 2), the most stable internal reference genes in different tissues under multiple-stress conditions could be selected (Table 3). *CACS* and *PP2A* were identified to be two stable reference genes in roots under all three stress-treatments (WR sample). Moreover, *CACS* was also stable as a reference gene in roots under phosphate deficiency (PR sample), aluminum toxicity (AR sample), or multiple stresses of aluminum toxicity and phosphate deficiency (MR sample). In stems under all three stress treatments (WS sample), *GAPDH* and *EIF4α* were found to be the most stable reference genes. *TIP41* and *ACT* were relatively stable for the PS sample, and *GAPDH* was the most stable reference gene in the MS sample. *GAPDH*, *CACS*, *PP2A* and *EIF4α* were four stable reference genes in the AS sample. For leaves in all three stress treatments (WL sample), *PP2A* and *TIP41* were shown to be the two most stable reference genes. *GAPDH* was relatively stable in the PL and AL samples, *TIP41* was relatively stable in the WL, PL and ML samples, and *HNR* was relatively stable in the AL sample. In addition, *CACS* was the most stable reference gene in the ML sample.

In summary, this study is the first to provide a group of reliable reference genes and use them to quantify the expression levels of functional genes of *Eremochloa ophiuroides* under multiple-stress conditions of P deficiency and/or aluminum toxicity.

## Figures and Tables

**Figure 1 plants-12-03751-f001:**
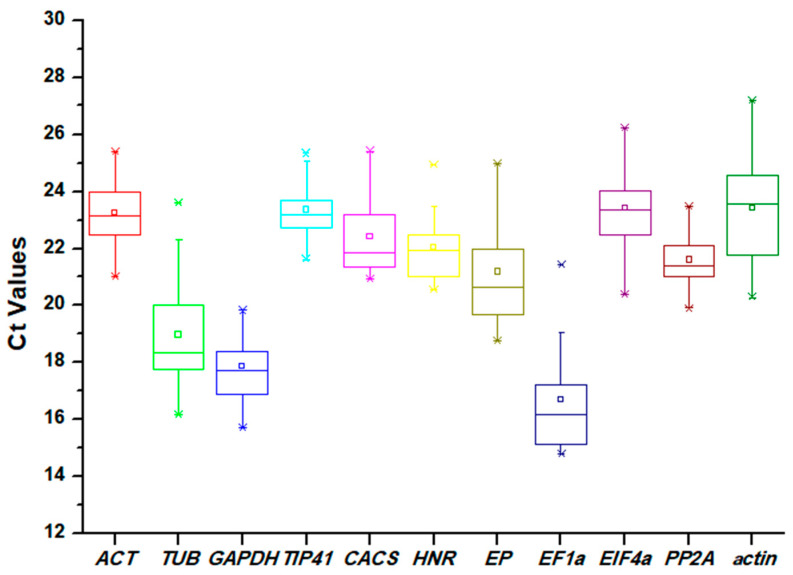
Distribution of Ct values of 11 candidate reference genes acquired by qRT-PCR in *Eremochloa ophiuroides*. Ct values for each reference gene are tested in all samples. The boxes indicate the 25th and 75th percentiles, with the lines in the center of the boxes representing the medians. The whiskers and asterisks represent the 99% confidence intervals and outliers, respectively. The upper and lower horizontal lines indicate the maximum and minimum values, respectively, and the small squares represent the average values.

**Figure 2 plants-12-03751-f002:**
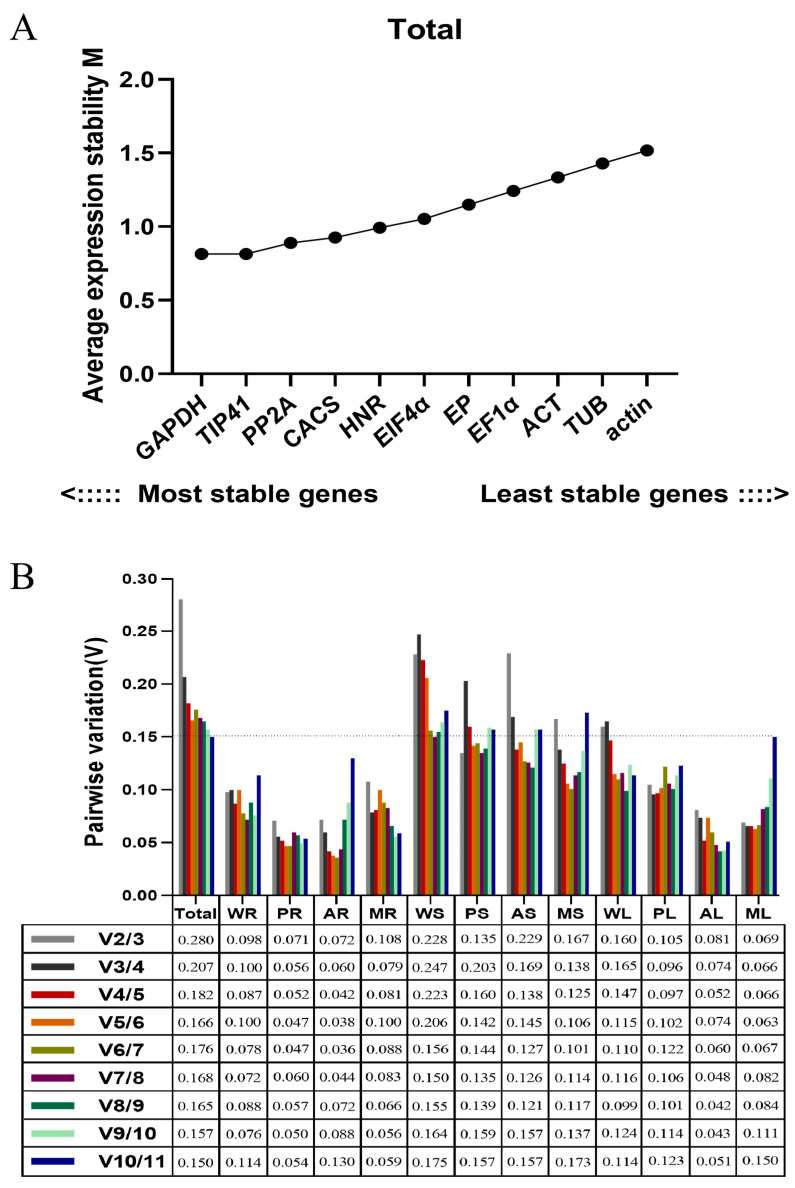
Average expression stability values and pairwise variations determined with geNorm_v3.5 software. (**A**) The expression stability values (M) and rankings of the 11 candidate reference genes of *Eremochloa ophiuroides.* (**B**) Optimal number of reference genes required for normalization in diverse samples. Total: all samples pooled; WR: pooled whole samples of root from all treatments; PR: P deficiency-treated roots; AR: Al-toxicity-treated roots; MR: Al-toxicity and P-deficiency multi-stress-treated roots; WS: pooled stem samples from all treatments; PS: P-deficiency-treated stems; AS: Al-toxicity-treated stems; MS: Al-toxicity and P-deficiency multi-stress-treated stems; WL: pooled leaf samples from all treatments; PL: P-deficiency-treated leaves; AL: Al-toxicity-treated leaves; ML: Al-toxicity and P-deficiency multi-stress-treated leaves.

**Figure 3 plants-12-03751-f003:**
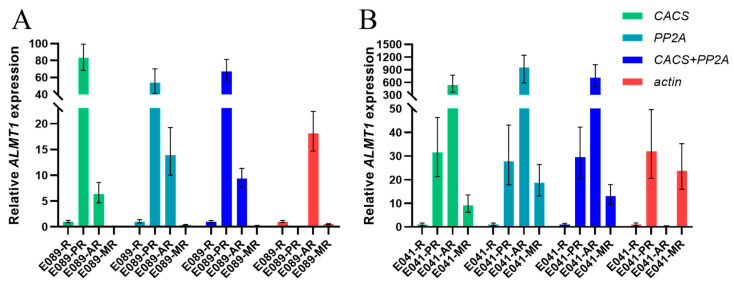
Validation of the most and least stable reference genes for the normalization of gene expression in *Eremochloa ophiuroides*. (**A**): Validation in the acid-soil-sensitive strain ‘E089’; (**B**): Validation in the acid-soil-resistant strain ‘E041’. R: Normal roots; PR: P-deficiency-treated roots; AR: Al-toxicity-treated roots; MR: Al-toxicity and P-deficiency multi-stress-treated roots.

**Table 1 plants-12-03751-t001:** qRT-PCR primers of candidate reference genes in *Eremochloa ophiuroides*.

Gene	Primer Sequence (5’ to 3’)	Tm (°C)	Amplification Size (bp)	Mean Efficiency	R
*ACT*	TGCATACGTTGCGCTCGACTA	62.8	155	1.863	0.9999
	CGATGAAGGATGGCTGGAAGA	62.3			
*TUB*	GCTTTCCTCGTATGCTCCTGTG	61.6	220	1.880	0.9995
	GATGGTGCGCTTGGTCTTGAT	62.3			
*GAPDH*	TTGAAGGGTGGTGCCAAGAAG	62.4	120	1.843	0.9999
	AGCAAGAGGAGCAAGGCAGTT	60.9			
*TIP41*	ATACTGTGGGAGTGATGCTGTG	57.4	157	1.852	0.9997
	CAAGATAACCTCGTCGTAGAAA	54.6			
*CACS*	TAAGATGTGATGTGACGGGAAAG	59.4	226	1.859	0.9997
	TCTGGTGGCACGAAACTGACT	60.3			
*HNR*	GTTGTTGATCGTGCGACTCCG	63.6	204	1.931	0.9998
	TATCTTCTTGCCCACTGCTTG	58.2			
*EP*	GAGACCGATCTCAACGAGGCT	60.7	223	2.000	0.9999
	TGCGCTTGGTGACATACATTAGG	62.7			
*EF1α*	GGATCTGAAGCGTGGGTATGT	58.9	239	1.868	0.9999
	CACCGTTCTTGAGGAATTTGG	59.7			
*EIF4α*	GACTATTTGGGTGTCAAAGTGC	56.6	207	1.874	0.9997
	ATCCTTGAATCCACGGGAAAG	60.5			
*PP2A*	ATTCAACCATACAAATGGGCTAAG	59.8	194	1.867	0.9996
	CTGGGTCGAATTGGAGGAAGT	60.5			
*actin*	GCACGGAATCGTCAGCAA	58.1	288	1.923	0.9934
	CCCTCGTAGATGGGGACAGT	58.6			

**Table 2 plants-12-03751-t002:** Stabilities of candidate reference genes ranked by RefFinder.

Rank	1	2	3	4	5	6	7	8	9	10	11
Total	*GAPDH*	*TIP41*	*HNR*	*PP2A*	*CACS*	*EIF4α*	*ACT*	*EP*	*EF1α*	*TUB*	*actin*
WR	*CACS*	*PP2A*	*EP*	*EF1α*	*GAPDH*	*TIP41*	*EIF4α*	*HNR*	*ACT*	*TUB*	*actin*
PR	*EF1α*	*CACS*	*EP*	*GAPDH*	*TIP41*	*HNR*	*PP2A*	*EIF4α*	*TUB*	*ACT*	*actin*
AR	*CACS*	*EF1α*	*EIF4α*	*PP2A*	*TIP41*	*TUB*	*GAPDH*	*HNR*	*EP*	*ACT*	*actin*
MR	*CACS*	*EP*	*PP2A*	*ACT*	*TUB*	*GAPDH*	*TIP41*	*EF1α*	*EIF4α*	*actin*	*HNR*
WS	*GAPDH*	*EIF4α*	*HNR*	*EP*	*CACS*	*actin*	*ACT*	*PP2A*	*TIP41*	*EF1α*	*TUB*
PS	*TIP41*	*ACT*	*GAPDH*	*EIF4α*	*PP2A*	*EP*	*HNR*	*CACS*	*actin*	*EF1α*	*TUB*
AS	*GAPDH*	*CACS*	*PP2A*	*EIF4α*	*ACT*	*TIP41*	*HNR*	*actin*	*EP*	*EF1α*	*TUB*
MS	*GAPDH*	*ACT*	*TIP41*	*HNR*	*EP*	*actin*	*CACS*	*EIF4α*	*EF1α*	*PP2A*	*TUB*
WL	*PP2A*	*TIP41*	*HNR*	*GAPDH*	*CACS*	*EP*	*ACT*	*TUB*	*EIF4α*	*actin*	*EF1α*
PL	*GAPDH*	*TIP41*	*HNR*	*PP2A*	*TUB*	*EIF4α*	*CACS*	*ACT*	*EP*	*EF1α*	*actin*
AL	*GAPDH*	*HNR*	*TIP41*	*CACS*	*TUB*	*actin*	*ACT*	*EF1α*	*EP*	*PP2A*	*EIF4α*
ML	*CACS*	*TIP41*	*EP*	*actin*	*ACT*	*PP2A*	*GAPDH*	*HNR*	*EIF4α*	*TUB*	*EF1α*

**Table 3 plants-12-03751-t003:** Most stable and least stable combinations of reference genes determined by comprehensive analysis.

WR		PR		AR		MR	
Most	Least	Most	Least	Most	Least	Most	Least
*CACS*	*actin*	*EF1α*	*actin*	*CACS*	*actin*	*CACS*	*HNR*
*PP2A*		*CACS*		*EF1α*		*EP*	
WS		PS		AS		MS	
Most	Least	Most	Least	Most	Least	Most	Least
*GAPDH*	*TUB*	*TIP41*	*TUB*	*GAPDH*	*TUB*	*GAPDH*	*TUB*
*EIF4α*		*ACT*		*CACS*		*ACT*	
				*PP2A*		*TIP41*	
				*EIF4α*			
WL		PL		AL		ML	
Most	Least	Most	Least	Most	Least	Most	Least
*PP2A*	*EF1α*	*GAPDH*	*actin*	*GAPDH*	*EIF4α*	*CACS*	*EF1α*
*TIP41*		*TIP41*		*HNR*		*TIP41*	

## Data Availability

The sequence data of the genes have been submitted to the GenBank and acquired the accession numbers (*CACS*: OR493130; *PP2A*: OR493131; *GAPDH*: OR493132; *EIF4α*: OR493133; *TIP41*: OR493134; *EF1α*: OR493135; *ACT*: OR493136; *HNR*: OR493137; *EP*: OR493138; *TUB*: OR493139).

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
