# Peer review of "Identification of Accurate Reference Genes for qRT-PCR Analysis of Gene Expression in Eremochloa ophiuroides under Multiple Stresses of Phosphorus Deficiency and/or Aluminum Toxicity"

_plants, 2023, doi:10.3390/plants12213751_

Round 1
Reviewer 1 Report
Comments and Suggestions for Authors
1, The identified reference genes should be identified in dicotyledon Arabidopsis and monocotyledon rice to determine or verify whether there were differences. Can it be used universally?
2, Is it necessary to select different reference genes for different tissue parts of the same plant in qPCR? Please provide further explanation or experimental verification.
3, In this study, four internal stability analysis software programs were selected to comprehensively analyze the accuracy of internal reference genes in expression analysis. It is suggested to simplify this part of the content or select some of the options for analysis.
4, There are too many Tables in the text. It is recommended that Table 2,3,4,5 be attached.
5, The title of the paper is unclear and should be revised.
Comments on the Quality of English LanguageModerate editing of English language required.
Reviewer 2 Report
Comments and Suggestions for Authors
The manuscript presented expression levels of potential reference genes for centipedegrass were studied. The work importance is of no doubt as the plant is supposed as well adapted for Al contaminated and nutrition deficient soil. The search of stable reference genes is significant for every other investigation for quantification the expression level of any functional genes.
Corrects should be made for line 56.
Reviewer 3 Report
Comments and Suggestions for Authors
The manuscript titled "Reference genes for qRT-PCR Analysis of Gene Expression in Eremochloa ophiuroides under Phosphorous Deficiency and Aluminum Toxicity Stress" gives detailed information about reference genes to be used for qRT-PCR for centipedegrass under different stress condition and different tissues.
A few information is missing which needs to be included:
1. Line 30 and 31: do write the full form of all the candidate reference genes
2. Line 106 and Fig S1: label the wells or the bands both for marker/ladder and the PCR product; 11 candidate reference genes are mentioned but only 10 genes are shown in the gel, do include the one missing (assuming actin is missing)
3. Line 111 and Table 1: include one more column which has Tm for each primer
4. Line 307: what temperatures were the plants grown in? as this can have effect on expression of the candidate reference genes
Round 2
Reviewer 1 Report
Comments and Suggestions for Authors
The author has answered my question and there are no further questions.
Comments on the Quality of English LanguageModerate editing of English language required
Reviewer 3 Report
Comments and Suggestions for Authors
The revised manuscript looks good. Thanks for working on it.